

**Single-particle characterization of aerosols collected at a remote site in the Amazonian**
**rainforest and an urban site in Manaus, Brazil**
Li Wu[1], Xue Li[1], HyeKyeong Kim[1], Hong Geng[2], Ricardo H. M. Godoi[3], Cybelli G. G. Barbosa[3],
Ana F. L. Godoi[3], Carlos I. Yamamoto[4], Rodrigo A. F. de Souza[5], Christopher Pöhlker[6], Meinrat
O. Andreae[6,7], and Chul-Un Ro[1,*]
[1] Department of Chemistry, Inha University, Incheon, 402-751, Korea;
[2] Institute of Environmental Science, Shanxi University, Taiyuan 030006, China;
[3] Environmental Engineering Department, Federal University of Parana-UFPR, Curitiba, PR,
Brazil;
[4] Chemical Engineering Department, Federal University of Paraná-UFPR, Curitiba, PR, Brazil;
[5] Amazonas State University, Superior School of Technology, Manaus, Amazonas, Brazil;
[6] Multiphase Chemistry & Biogeochemistry Departments, Max Planck Institute for Chemistry,
55020 Mainz, Germany;
[7] Department of Geology and Geophysics, King Saud University, Riyadh, Saudi Arabia.
**Abstract**
In this study, aerosol samples collected at a remote site in the Amazonian rainforest and an urban
site in Manaus, Brazil, were investigated on a single particle basis using a quantitative energy-
dispersive electron probe X-ray microanalysis (ED-EPMA). Twenty-three aerosol samples were
collected in four size ranges (0.25-0.5, 0.5-1.0, 1.0-2.0, and 2.0-4.0 µm) during the wet season in
2012 at two Amazon basin sites: 10 samples in Manaus, an urban area; and 13 samples at an 80-m
high tower, located at the Amazon Tall Tower Observatory (ATTO) site in the middle of the
rainforest, 150 km northeast of Manaus. The aerosol particles were classified into nine particle

*Corresponding author. Tel.: +82 32 860 7676; fax: +82 32 874 9207

E-mail address: curo@inha.ac.kr (C.-U. Ro)



types based on the morphology on the secondary electron images (SEIs) together with the elemental concentrations of 3,162 individual particles: (i) secondary organic aerosols (SOA), (ii) ammonium sulfate (AS), (iii) SOA and AS mixtures, (iv) aged mineral dust, (v) reacted sea-salts, (vi) primary biological aerosol (PBA), (vii) carbon-rich or elemental carbon (EC) particles, such as soot, tar ball, and char, (viii) fly ash, and (ix) heavy metal (HM, such as Fe, Zn, Ni, and Ti)-containing particles. In submicron aerosols collected at the ATTO site, SOA and AS mixture particles were predominant (50-94% in relative abundance) with SOA and ammonium sulfate comprising 73-100%. In supermicron aerosols at the ATTO site, aged mineral dust and sea-salts (37-70%) as well as SOA and ammonium sulfate (28-58%) were abundant. PBAs were observed abundantly in the $PM_{2-4}$ fraction (46%), and EC and fly ash particles were absent in all size fractions. The analysis of a bulk $PM_{0.25-0.5}$ aerosol sample from the ATTO site using Raman microspectrometry and attenuated total reflection Fourier transform infrared spectroscopy showed that ammonium sulfate, organics, and minerals are the major chemical species, which is consistent with the ED-EPMA results. In the submicron aerosols collected in Manaus, either SOA and ammonium sulfate (17-80%) or EC particles (6-78%) were dominant depending on the samples. In contrast, aged mineral dust, reacted sea-salt, PBA, SOA, ammonium sulfate, and EC particles comprised most of the supermicron aerosols collected in Manaus. The SOA, ammonium sulfate, and PBAs were mostly of a biogenic origin from the rainforest, whereas the EC and HM-containing particles were of an anthropogenic origin. Aged mineral dust and reacted sea-salt particles, including mineral dust mixed with sea-salts probably during long-range transatlantic transport, were abundant in the supermicron fractions at both sites. Among the aged mineral dust and reacted sea-salt particles, sulfate-containing ones outnumbered those containing nitrates and sulfate+nitrate in the ATTO samples. In contrast, particles containing sulfate+nitrate were comparable in number to particles containing sulfate only in the Manaus samples, indicating the different sources and formation mechanisms of secondary aerosols, i.e., the predominant presence of sulfate at the ATTO site from mostly biogenic emissions and the elevated influences of nitrates from anthropogenic activities at the Manaus site.

## 1. Introduction

The Amazonian rainforest is regarded as one of the primitive continental regions and atmospheric aerosol particles over the region are expected to be influenced minimally by



anthropogenic activities, particularly during the wet season (Andreae, 2007; Martin et al., 2010b;
Chen et al., 2015). The unique near-natural conditions during the wet season make it an ideal place
to understand the occurrence, nature, origin, and transport of aerosol particles, which can directly
scatter and absorb solar radiation and indirectly serve as cloud condensation nuclei (CCN) and/or
ice nuclei (IN), to better predict the additional anthropogenic effects on aerosol particles, and to
help determine their influences on the environment, climate, and human health (Artaxo et al.,

2013).

Many studies have been performed on the aerosol characteristics in the Amazon basin, but

the formation and interaction of airborne Amazonian aerosols are not completely understood
(Andreae et al., 2015; Martin et al., 2010a, 2016; Fraund et al., 2017; Fan et al., 2018). The Amazon
Basin atmosphere is near-pristine during the wet season, whereas biomass burning prevails during
the dry season (Andreae et al., 2007; Pöschl et al., 2010; Artaxo et al., 2013; Pöhlker et al., 2018).
Based on a long-term study, it was reported that aerosol particles in the coarse fraction are
relatively constant in concentrations through the wet and dry seasons, whereas the aerosol particle
levels in the fine fraction differ due to the predominant influence of biomass burning during the
dry season (Artaxo et al., 2013; Moran-Zuloaga et al., 2018). Scanning electron microscopy/energy
dispersive X-ray spectrometry (SEM/EDX) studies categorized the Amazonian aerosols mainly as
secondary organic aerosol (SOA) particles, sulfates/chlorides, primary biological aerosol (PBA)
particles, mineral dust, sea salts (fresh and/or aged), and pyrogenic carbon particles within the
different size fractions (Krejci et al., 2005; Pöschl et al., 2010). Over the Amazonian rainforest,
SOA particles are formed through the condensation of biogenic organic compounds onto biogenic
K-rich salt particles emitted from the forest and are predominant in the fine fraction, which are
important for CCN (Pöhlker et al., 2012) and can also affect the potential of mineral particles when
acting as an organic coating (Möhler et al., 2008). Under high relative humidity conditions, nano-
and micrometer SOA particles with a dominance of α-pinene and isoprene as their precursors can
remain in the liquid phase (Bateman et al., 2016), which further enhances the formation of SOA
as well as the oxygen-to-carbon (O/C) ratios. Hence, the study of this particle type can help
elucidate some of the atmospheric interactions (Lin et al., 2014). In addition, the hygroscopicity
of complex ambient organic particles from the Amazon Basin was well simulated by a mixture of
organic surrogates consisting of levoglucosan, 4-hydroxybenzoic acid, and humic acid (Lei et al.,
2018). The atmosphere in the Amazon Basin is also rich in PBA particles (Andreae, 2007; Artaxo



et al., 1998, 2013; Martin et al., 2010a). Their unique morphology and elemental compositions of
major C and O with minor S, K, P, Na, N, Cl, and/or Mg obtained by SEM/EDX are characteristic
of individual PBAs like fungal spores (China et al., 2016). PBA particles can contribute to CCN
after being transported to cloud formation altitudes by strong convection (Artaxo et al., 2013).
The Amazon Tall Tower Observatory (ATTO) consists of several observatory towers built
in the middle of the Amazon rainforest for a continuous and detailed study of biota-atmosphere
interactions (Andreae et al., 2015). At the ATTO site, single particle analysis by a combination of
scanning transmission X-ray microscopy/near edge X-ray absorption fine structure spectroscopy
(STXM/NEXAFS) and SEM/EDX highlighted the dominance of biological particles (Fraund et
al., 2017). Very few cluster varieties were observed without anthropogenic elements that were
found near a capital city, Manaus. The abundance of biogenic SOA and the presence of C, N, O, P
and K are characteristic of aerosols at the area (Fraund et al., 2017).
Manaus, the capital of Amazonas state, is a large city located in the northern region of
Brazil with more than 2 million inhabitants in an area of 11,401 km² (IBGE, 2017). The city, which
is surrounded by the largest tropical rainforest, has a large industrial zone, a port area at the Rio
Negro, an energy matrix based on fuel oil, diesel, and natural gas, and a growing automotive fleet
(Martin et al, 2010a, 2016). Consequently, the pollution plume from Manaus can act as a laboratory
for examining the perturbations in natural processes (Martin et al, 2016). Based on an investigation
on particulate matter during the wet season, more oxidized organic components were observed to
be present at sites downwind of Manaus than the upwind ones (de Sá et al., 2018), of which one
third was of an urban origin (Palm et al., 2018).
Only a few studies examined airborne particles over the Amazon rainforest and nearby
urban sites simultaneously (Fraund et al., 2017; Martin et al., 2016). Therefore, there is still little
information on the urban vs. ecosystem influences. In this study, twenty-three aerosol samples
collected at the ATTO site and at an urban site in Manaus during the wet season in 2012 were
examined on a single particle basis using a quantitative energy-dispersive electron probe X-ray
microanalysis (ED-EPMA), which provided information on the morphology and chemical
compositions of aerosols containing both light and heavy elements. This paper presents the
different characteristics of the aerosols collected at the rainforest and in Manaus.

**2. Experimental section**





## 2.1. Samples

During the wet season in 2012, aerosol samples were collected at two sampling sites on the Amazon basin, i.e., ATTO and a central area of Manaus. The ATTO site (S 02° 08.647' W 58° 59.992') is situated in the Uatumã Sustainable Development Reserve, approximately 150 km northeast of Manaus (Fig. 1). This is a multidisciplinary research site of an international joint project between Brazil and Germany for continuous monitoring of the biological, physical, and chemical functions of the Amazon rainforest to answer questions related to climate change (Andreae et al., 2015). Aerosol sampling was performed at an 80-m-height walk-up tower at the ATTO site. In Manaus, the sampling site is situated in the central part of the city (S 03° 05.753', W 59° 59.419'), a representative urban site influenced mostly by vehicle traffic. Particles were collected at a 2 m height above ground level. The aerosol samples were collected on TEM grids (Ted Pella Inc., USA, Ted Pella Inc., Carbon/Formvar 200 mesh Cu grid, 35−70 nm thickness) using a five stage Battelle impactor (the cut-off diameters are 0.25, 0.5, 1, 2, and 4 μm for stages 1-5, respectively) at the ATTO and Manaus sites on April 1, 16, 17, and 18 and May 1, 2, and 3 (the four- and three-day samples were collected at the ATTO and Manaus sites, which are notated as samples SA1-SA4 and SM1-SM3, respectively). On each date, the sampling started around noon (local time) and lasted for approximately 100 min. The individual particles collected on stages 1-3 ($PM_{0.25–0.5}$, $PM_{0.5–1.0}$, and $PM_{1.0–2.0}$) for each sample and on an additional stage 4 ($PM_{2.0–4.0}$) for the SA4 and SM3 samples were examined.

During the sampling period, the temperature was in the range of 22 to 32°C and the relative humidity was above 55%. On April 16 and May 1 and 2, rain events occurred within the previous 24 hours prior to sampling. The ten-day backward air-mass trajectories were obtained using the Hybrid Lagrangian Single-Particle Integrated Trajectory (HYSPLIT) model from the NOAA Air Resources Laboratory's web server (http://www.arl.noaa.gov/ready/hysplit4.html), as shown in Fig. 2. All samples were influenced by transatlantic air masses at a 1000 m receptor height and the Manaus site was influenced mainly by the surrounding rainforest at 500 m and 100 m heights.

## 2.2. EPMA measurements and data analysis

Low-$Z$ particle EPMA was carried out by SEM (JSM-6390, JEOL) equipped with an Oxford Link SATW ultrathin window EDX detector, which has a spectral resolution of 133 eV for Mn $K_\alpha$ X-rays. The X-ray spectra were recorded using INCA Energy software. To achieve the





optimal experimental conditions, such as the low background level in the X-ray spectra and good
sensitivity for low-$Z$ element analysis, an accelerating voltage of 10 kV, a beam current of 0.5 nA,
and a measuring time of 20 s were used. X-ray spectral data acquisition for individual particles
was carried out manually in point analysis mode, i.e., the electron beam was focused at the center
of each particle, and X-rays were acquired while the beam remained fixed on this single spot. The
secondary electron images (SEIs) and X-ray spectra of an overall 3,162 individual particles for the
ATTO and Manaus samples were examined. As the TEM grids are thin (35-70 nm thickness),
strong X-rays from the Al or Cu metal stub commonly used in the SEM/EDX measurement would
be a problem when the TEM grid substrate is placed on it. A home-made sample holder (Fig. 3(a))
for the TEM grid samples was used to avoid interference from the metal stub, resulting in X-ray
spectra of bare TEM grids, which showed only C and O X-ray peaks from their carbon/Formvar
thin-film, a Cu-L peak caused by lateral scattering from the Cu bars of TEM grids, and a Si peak
from an impurity, as shown in Fig. 3(b). The net X-ray intensities for the chemical elements were
obtained by non-linear, least-square fitting of the spectra using the AXIL program (Vekemans et
al., 1994). Although the characteristic X-ray intensities of C and O were low for the bare TEM
grids, determination of the C and O concentrations for individual particles on the TEM grids was
performed using a methodology based on the Monte Carlo calculation technique to correct for the
interfering X-ray peaks of C and O emitted from the TEM grid, which provided reliable
quantification results when applied to the quantification of standard (sub)micron particles, such as
$CaCO_3$, $CaSO_4$, $Na_2SO_4$, and $SiO_2$. On the other hand, electron beam-sensitive particles, such as
$NaNO_3$, $Ca(NO_3)_2 \cdot 4H_2O$, and ammonium sulfate, provided deviating quantification results (Geng
et al, 2010). As the Cu-L and Si X-ray intensities from the bare TEM grids are quite small (< 20
cps) under these measurement conditions, the two peaks could be neglected safely during the
quantification procedure.

**3. Results and discussion**
**3.1. Particle types observed in samples collected at the ATTO and Manaus sites**

In this study, the analyzed particles were classified based on their X-ray spectral and SEI

data, where 9 different particle types were observed in the samples collected at the ATTO and
Manaus sites during the wet season in 2012; i.e., (i) SOA, (ii) ammonium sulfate (AS) particles,
(iii) SOA and AS mixture particles, (iv) aged (reacted) sea-salt, (v) aged mineral dust, (vi) PBA



particles, (vii) carbon-rich or elemental carbon (EC) particles such as soot, tar ball, and char or
coal dust, (viii) fly ash particles, and (ix) heavy metal-containing (HM) particles. Based on single
particle analysis for aerosol samples collected at a remote site north of Manaus, Brazil during the
2008 wet season (3-13 March) using SEM/EDX, five types of aerosols, such as (i) SOA droplets,
(ii) SOA-inorganic mixture particles where the inorganics are mostly sulfates and/or chlorides, (iii)
PBA, (iv) mineral dust, and (v) pyrogenic carbon particles, were reported. The pure SOA droplets
dominated in the nucleation and Aitken modes, whereas the pure SOA, SOA-inorganic mixture
particles, and pyrogenic carbon particles dominated in accumulation mode (Pöschl, et al., 2010).
With the exception of the reacted sea-salt particles probably from the Atlantic Ocean, the particle
types observed in this study are comparable to their study. Figs. 4 and 5 present typical field SEIs
for submicron and supermicron aerosol particles collected at the ATTO and Manaus sites,
respectively, where the chemical species comprising each particle is indicated. Ammonium sulfate
and SOA particles are dominant in the sub- and super-micron aerosol fractions collected at the
ATTO site with some mineral particles and aged sea-salts in the supermicron fractions, whereas
the aerosol samples collected at the Manaus site are composed of various types of particles of
anthropogenic and/or natural origin. Figs. S1-S7 of Supporting Information present typical SEIs
for all the samples with identified chemical species on the SEIs, which helps briefly illustrate the
different features of the samples collected at the ATTO and Manaus sites. The characteristics of
the particle types observed in the ATTO and Manaus samples are described in the following.

### 201 3.1.1. Secondary organic aerosol (SOA) particles

In this study, SOA particles were observed frequently in both the ATTO and Manaus
samples, even though pure SOA particles were rare and most of them were mixed internally with
other species, such as ammonium sulfate, K-rich salt, reacted sea-salts, etc. The SOA particles over
the Amazon rainforest are formed by the oxidation of biogenic volatile organic compounds
(Jimenez et al., 2009; Hallquist et al., 2009; Martin et al., 2010a) and are the major constituents of
particulate matter (PM), particularly for submicron ambient PM (Pöschl, et al., 2010; Martin et al.,
2010b; Chen et al., 2015). In the SEI images, pure SOA droplet particles appear gray in contrast
and have a circular shape, as shown in Fig. 6(a). As TEM grid films (with 90% C content) and
SOAs are composed mainly of carbon and oxygen, the SOA aerosols appear gray on the TEM
grids because of their similar secondary and backscattered electron yields to those of the TEM grid



(Goldstein et al., 2003; Maskey et al., 2010). As TEM grids are hydrophobic due to the thin carbon
layer over the Formvar film, the aqueous droplet aerosols appear circular on the TEM grids (Eom
et al., 2014; Maskey et al., 2010), suggesting that SOAs were collected as aqueous droplets at the
time of particle sample collection. Recent studies also reported that most submicron SOA particles
in the Amazon basin are water soluble organic aerosols (WSOAs) rather than semi-solid or solid
aerosols under the background conditions that are typically met during the wet season (Bateman
et al., 2016, 2017). The X-ray spectrum of a typical pure SOA, as shown in Fig. 6(a), showed
considerably higher levels of the C X-ray peak intensity compared to that from the Formvar/carbon
film of the TEM grids, resulting in the unambiguous identification of SOA particles based on their
SEIs and X-ray spectral data.

### 3.1.2. Ammonium sulfate (AS) particles

Ammonium sulfate particles were observed abundantly in the ATTO and Manaus samples,
mostly as mixtures with secondary organics. Ammonium sulfate particles appear bright and
crystalline on the SEIs before the X-ray measurements, both for pure airborne and standard
ammonium sulfate particles, as shown in Figs. 6(b) and 6(f), respectively. The standard ammonium
sulfate particles were deposited on TEM grids by the nebulization of a 0.1 M ammonium sulfate
solution. As shown in the inset in Figs. 6(b) and 6(f), after the X-ray measurements, they show
somewhat darkened SEIs with black holes, due to electron beam damage, at the places where the
electron beam hits. As the ammonium sulfate particles are electron beam-sensitive (Geng et al.,
2010; Worobiec et al., 2003; Huang and Turpin, 1996), their X-ray spectral signature is the
presence of a significant S X-ray peak, as shown for both pure airborne and standard particles on
the TEM grids. The N X-ray peak was often not detected, particularly for small particles because
the $NH_4^+$ moiety is especially prone to damage by electron beams.
Ambient urban and rural sulfates act as a sink for ammonia, of which the sources are largely
animal waste, fertilizer application, soil release, and industrial emissions. The most common form
is ammonium sulfate. On the other hand, if ammonia is scarce in the air, sulfates would be in more
acidic forms, such as $NH_4HSO_4$ or $H_2SO_4$ (Millstein et al., 2008). The acidic $NH_4HSO_4/H_2SO_4$
particles have been reported to be more hygroscopic than pure ammonium sulfate (Pósfai et al.,
1998). Hence, they can be spread over the collecting substrate (Formvar/carbon film). In addition,
acidic sulfate particles can have unique halo rings in their morphology (Buseck and Pósfai, 1999).



The crystalline structure of the ammonium sulfate-containing particles observed in this study suggests that they are sulfates fully neutralized with ammonia. In addition, the Raman spectra of airborne particles exhibiting this morphology were obtained on a single particle basis to confirm that they are ammonium sulfate. As shown in Fig. 7(a), the Raman peak at 975 cm$^{-1}$ of the airborne particle is characteristic of ammonium sulfate (Ling and Chan, 2007), which was also confirmed by Raman spectroscopy on standard ammonium sulfate particles. Characteristic Raman peaks for $NH_4HSO_4$, $K_2SO_4$, $CaSO_4 \cdot 2H_2O$, $CaSO_4$, $Na_2SO_4$, and $MgSO_4 \cdot xH_2O$ (x = 1-7, 11) were reported to be at 1010 and 1042, 983, 1008, 1014 and 1025, 992, and 984-1046 cm$^{-1}$, respectively (Fung and Tang, 1988; Wang et al., 2006; Mabrouk et al., 2013; Prieto-Taboada et al., 2014). The sloping baseline in the Raman spectrum of the airborne particle was attributed to the fluorescence from organics, indicating the presence of organic compounds (Sobanska et al., 2012), probably from SOA. For aerosols collected on Ag foil at the ATTO site on June 10, 2014, ammonium sulfate is the major species with some organics and minerals such as kaolinite, for the bulk aerosols in the size range of 0.25 – 0.5 µm, based on their X-ray, attenuated total reflectance-FTIR (ATR-FTIR), and Raman spectra (Fig. 7(b)). A study of the samples collected in the central Amazon Basin during the wet season from February to March 2008 reported that ammonium was not sufficient to fully neutralize sulfates so that ammonium bisulfate would be present in the Amazon rainforest (Chen et al, 2015), whereas other studies reported that sulfates are sufficiently neutralized with ammonia in the fine and coarse fractions during both the wet and dry seasons (Andreae et al., 2015; Martin et al., 2010b; Fuzzi et al., 2007; Mace at al., 2003). The different results may be due to different sampling places and seasons. In this study, ammonium sulfate is dominant over ammonium bisulfate.

Previous studies have shown that the sulfate aerosols over the Amazon forest are predominantly from marine and terrestrial biogenic sources, with comparable contributions from marine and terrestrial biogenic emissions (Andreae et al., 1990). Sulfate originates from biogenic sources in the rainforest, i.e., dimethyl sulfide (DMS), $H_2S$, and $CS_2$ emitted by plants and microorganisms, which can be oxidized to sulfate (Andreae et al., 1990, 2015; Martin et al., 2010b). The rainforest ecosystem in the central Amazon can act as a source of DMS to the atmosphere throughout the year (Jardine et al., 2015). Several studies have reported that marine DMS transported from the Atlantic Ocean contributes significantly to the sulfate levels in the Amazon basin (Gregory et al., 1986; Andreae et al., 1990; Martin et al., 2010a). In addition, there is some



sulfate from long-range transport across the Atlantic and minor upwind anthropogenic sources.
The nitrogen cycle is essential for organisms and some bacteria to fix the gaseous $N_2$ in the
air to $NH_4^+$ for their own biosynthetic processes (Kim and Rees, 1994; Bazzaz, 1998; Kellerhals
et al., 2010). In addition, some microorganisms produce enzymes to release nitrogen as $NH_4^+$
during the nitrogen mineralization process, which is important in tropical rain forest soils, where
dead plants and animal matter accumulate continuously (Wright, 1996; Neill et al., 1999). A high
level of $NH_4^+$ in tropical rain contributes significantly to the nitrogen influx in the rainforest soils
(Jordan et al., 1982). The $NH_4^+$ species can be evaporated as gaseous $NH_3$ from surface soils,
particularly from leaf litter, resulting in a strong $NH_3$ emission source as well as stomatal $NH_3$
emission of plants as another natural source in forest ecosystems (Sutton et al., 2009, 2013; Hansen
et al., 2017). On the other hand, $NH_4^+$ species in rainforest soils might become airborne
immediately after rainfall, similar to the way that airborne organic particles are produced directly
from soils by raindrop impaction (Wang et al., 2016). The wetness of forest surfaces is significant
in controlling both the deposition and emission of atmospheric $NH_3$ (Hansen et al., 2015). As
ammonium sulfate-containing particles were also observed abundantly in the samples collected at
Manaus site, they were influenced strongly by the surrounding Amazonian forest and/or generated
by anthropogenic activities in the urban environment.

### 3.1.3. SOA and AS mixture particles

In this study, most airborne submicron SOAs were observed to be internally mixed with
ammonium sulfate, particularly for the samples collected at the ATTO site. Figure 6(c) shows that
a typical SOA and AS mixture particle has the crystalline, bright ammonium sulfate moiety in the
center surrounded by circular, grey organic species. The circular morphology of the organic species
strongly suggests that the organic species are SOAs, as stated above. As efflorescence and
deliquescence relative humidity (ERH and DRH) of ammonium sulfate species are 30-40% and
80%, respectively (Yeung and Chan, 2010) and the ambient RH was always above 55% during
sampling for the ATTO and Manaus samples, the ammonium sulfate would be mostly in aqueous
droplets at the time of sample collection, rather than in crystalline form, as indicated by their
overall circular shape. When the particle samples were under dry conditions either during sample
storage or in the vacuum chamber of the SEM instrument, the ammonium sulfate species
crystallized, resulting in core-shell structures of organic and inorganic mixture aerosols. When the



ambient RH is low enough to make the ammonium sulfate species crystallize in the atmosphere,
the organic and inorganic mixture aerosols would similarly be present as core-shell structures.
Some of the SOA and AS mixture particles were also mixed with K-salts. As shown in Figs.
6(d) and 6(e), their morphology was similar to that of the SOA and AS mixture particles, but their
X-ray spectra revealed the presence of K and an elevated S level compared to those of the SOA
and AS mixture particles, suggesting that the K is associated mostly with $SO_4^{2-}$. The shoulder
Raman peak of the airborne ammonium sulfate particle at 982 cm$^{-1}$ (Fig. 7(a)), which is indicative
of the $K_2SO_4$ moiety (Mabrouk et al., 2013), also suggests that the K-salts are most probably $K_2SO_4$.
A previous study reported that small K-salt-rich particles can act as seeds for SOA formation in
the Amazon basin and K-salts are present ubiquitously in Amazonian SOAs with their content
being higher in the morning hours and for smaller SOAs (Pöhlker et al., 2012). On the other hand,
among the 843 submicron SOA and/or ammonium sulfate particles collected at the ATTO site,
only 31% contained K-salts, which is probably because the samples were collected in the afternoon
and/or the analyzed particles were larger than 0.25 μm so that the K-salt content may be below the
detection limit of EDX (~ 0.1 wt. %). In the Manaus samples, a total of 199 submicron SOA and/or
ammonium sulfate particles were observed, of which approximately 40% contained K-salts,
suggesting that the Manaus samples were influenced strongly by the surrounding rainforest as
supported by the backward trajectories (Figs. 2(e)-(g)), where K-salts may be mostly of a biogenic
origin in the rainforest. The organic moiety is often mixed internally with aged sea-salts, mineral
dust, and PBAs, which will be described below.

### 3.1.4. Mineral dust particles

The typical mineral dust particles include aluminosilicate, quartz ($SiO_2$), calcite ($CaCO_3$),
dolomite ($CaMg(CO_3)_2$), and $TiO_2$ (Geng et al., 2009, 2011). They appear irregular and bright on
the SEIs (Fig. 8). Various types of mineral particles from Saharan dust contribute significantly to
the nutrient cycles in the Amazon rainforest (Talbot et al., 1990; Abouchami et al., 2013; Rizzolo
et al., 2017). Mineral dust tends to provide reactive surfaces for heterogeneous reactions with trace
atmospheric gases, such as $SO_2$ and $NO_x$, leading to chemical modifications of the particles that
ultimately affect the atmospheric chemical balance and photochemical cycle (Sullivan et al., 2007;
Chen et al., 2011). Modification of the physicochemical properties of particles can alter their
optical, chemical, and hygroscopic properties (Sullivan et al., 2007; Geng et al., 2014). If some



components in them (particularly Ca-containing species) react with airborne $SO_2$ and $NO_x$ in the presence of moisture or with "secondary acids", such as $H_2SO_4$, $HNO_3$, and HCl, they are regarded as reacted or aged ones. The reacted/aged ones can contain either nitrates, sulfates, or both (Geng et al., 2014, 2017). In the Amazon samples, almost all the mineral dust particles were aged ones, as shown in Fig. 8, where an aluminosilicate and a carbonate/silica mixture particle containing sulfate are shown. The S-containing aged mineral dust particles outnumbered the N- and both N- and S-containing ones for the ATTO samples, whereas they were comparable to both the N- and S-containing ones for the Manaus samples, as shown in Fig. 9(a), indicating the predominance of sulfates over nitrates for the reaction of mineral dust particles at the ATTO site and somewhat significant influence of nitrates at the Manaus site. The mineral particles might be mixed or covered with SOA and/or ammonium sulfate and gradually become aged under a high RH in the rainforest. Some mineral particles were also mixed with sea salt particles during their transport to the Amazonian area across the Atlantic Ocean.

### 3.1.5. Reacted (aged) sea-salts

The nascent airborne sea-salt particles can react with gas-phase sulfur and nitrogen oxides to contain sulfate and nitrate, respectively (ten Brink, 1998). During the process, chlorine may be removed completely if the reaction is complete (Laskin et al., 2003). All sea-salt particles observed in the ATTO and Manaus samples were reacted ones. Figure 10(a) shows the X-ray spectrum, atomic concentration, and SEI of a typical aged sea salt particle, where the presence of Na, Mg, and Cl indicates its marine origin (Geng et al., 2014) and the presence of S and C indicates that it is mixed with sulfates and organics. The irregular and somewhat bright SEI is typical of the aged sea-salt particles. As shown in Fig. 9(b), S-containing sea-salts outnumbered N- and both N- and S-containing ones for the ATTO samples, indicating the predominance of sulfates over nitrates for the reaction of sea-salt particles. The S-containing particles are comparable in abundance to both the N- and S-containing ones at the Manaus site (Fig. 9(b)). The sea-salt particles may also become mixed with ammonium sulfate over the rainforest and become S-containing ones. Among overall 275 reacted sea-salts containing sulfate/nitrate and organics, 71% of them were mixed with K-salts, as shown in Fig. 10(b). The presence of K-salts in the reacted sea-salt particles indicates that the mixing of the K-salts of biogenic origin would happen in the rainforest during long-range transport because of the minimal biomass burning influence during the wet season. In addition, several



elongated CaSO$_4$ particles, as shown in Fig. 10(c), were detected in both ATTO and Manaus
samples, all of which contain a small amount of Na. Their elongated shape and the presence of Na
strongly indicates that they were from the sea, possibly the Atlantic Ocean, not from the soil (Eom
et al., 2016).

### 3.1.6. Primary biological aerosol (PBA) particles

PBA particles like fungal spores can be identified easily based on their unique morphology

and the presence of their characteristic chemical elements (Geng et al., 2009; Martin et al, 2010a;
Pöschl et al., 2010). The PBA particle shown in Fig. 11(a) has a unique oval morphology and the
majority of C together with the characteristic small amounts of N, P, S, Cl, and K. PBA particles
are relatively large (size > 2.0 μm) so that they are abundant in stage 4 samples (2.0 μm < size <
4.0 μm), particularly at the ATTO site. Figure 12 shows two image fields of stage 4 samples
collected at the ATTO and Manaus sites, where the PBA particles have various types of
morphology and many of them are mixed with SOA. The abundant observation of PBA particles
in the stage 4 sample of the Manaus site suggests the transport of the PBA particles from the
rainforest to the urban area. Supermicron PBA particles were reported to be abundant over the
Amazon (China et al., 2016; Moran-Zuloaga et al., 2017; Gilardoni et al., 2011). PBA particles can
be pollen, bacteria, fungal and fern spores, viruses, and fragments of plants and animals emitted
directly from the rainforest, showing a range of morphologies, and comprise the largest fraction of
the coarse mass (Martin et al, 2010a). PBA particles appear to be the most efficient and abundant
ice nuclei (Pöschl et al., 2010; Tobo et al., 2013; Haga et al., 2014). In addition, the release of
nano- and submicron particles from fungal spores under high relative humidity can contribute to
new particle formation and potentially affect cloud formation in the Amazon Basin (China et al.,

2016).


### 3.1.7. Carbon-rich particles from combustion sources

Carbon-rich particles, such as soot, tar balls, and char or coal dust, which contain more

than 90 at. % C and O with the C content being dominant over O in low-$Z$ particle EPMA analysis
(Geng et al., 2009, 2010, 2014), were observed frequently in the Manaus samples, whereas they
were rare in the ATTO samples. Based on the characteristic morphology of carbon-rich particles,
soot aggregates of fractal-like chain structures (Fig. 11(b)), tar balls of separate spherules (Fig.



11(c)), and chars of irregular-shaped carbon (Fig. S5, SM1-2) could be differentiated
straightforwardly from each other (Geng et al., 2010, 2014). The soot aggregates formed via a
vaporization-condensation mechanism during the combustion processes vary in size from sub to
several micrometers (Chen et al., 2005, 2006). Once airborne, the complex microstructure of the
soot aggregates may provide active sites for the deposition of organics and other chemical species,
such as sulfates (Pósfai et al., 1999; Zhang et al., 2008), as revealed by the presence of S in Fig.
11(b). This results in aged soot aggregates that become compact with considerable restructuring
and shrinkage (Zhang et al., 2008). Tar balls, which are a type of brown carbon, usually have high
C, N, and O contents with a spherical morphology (Fig. 11(c)), strongly indicating their formation
during biomass combustion processes (Pósfai et al., 2003, 2004). Char appears compact and
irregular in the SEIs; it is often mixed with minor inorganic species, such as K and S, and is
regarded as the incomplete combustion remnants of liquid or solid carbonaceous fuel materials
that have undergone carbonization during combustion (Chen et al., 2006). Only one soot particle
was observed in all the ATTO samples, whereas soot, tar ball, and char or coal dust particles were
abundant in the submicron Manaus samples, suggesting that the ATTO samples are barely affected
by the anthropogenic carbon-rich particles generated in Manaus.

### 3.1.8. Fly ash particles

As shown in Fig. 11(d), fly ash particles are glassy spheres, composed mainly of O, Si, and
Al with minor components, such as Fe and Ca, which can be identified clearly by their spherical
shape and bright contrast on SEIs. The fly ash particles are different from tar balls having only C
and O signals in their X-ray spectra, even though both are generated during the combustion
processes (Geng et al., 2017). They were observed only in the Manaus samples, reflecting their
anthropogenic origin.

### 3.1.9. Heavy metal-containing particles

Heavy metal-containing (HM) particles, such as Ni-, Ti-, Zn-, and Fe-containing ones,
appear bright and irregular on SEM images, as shown in Figs. 11(e) and (f), and were observed
mostly in the fine fraction with more than a half of them being Fe-containing particles both in the
ATTO and Manaus samples. The Fe-containing particles in the ATTO samples were observed to
be associated with SOA (and ammonium sulfate), as shown in Fig 11(f), indicating its mixing with



the species of a biogenic emission origin. Sahara mineral dust has been reported to be essential for the nutrient cycles in the Amazon rainforest because many types of minerals are carried and transported into the rainforest, in which Fe is one of the important micronutrients in a Fe-limited rainforest (Rizzolo et al., 2017). Among all the mineral dust particles observed in the samples, approximately 40% of them contain Fe. The HM particles can also be of anthropogenic origin: emitted from the streets or road surface as brake dust, road paint particles, diesel exhaust particles, construction materials, and/or car catalyst materials (Qiao et al., 2016).

**3.2. Relative abundances of particle types observed in the ATTO and Manaus samples**

Figure 13 shows the relative abundance of the nine different particle types observed in the ATTO and Manaus samples. In the stage 1 samples (0.25-0.5 μm size) of SA1-SA4 collected at the ATTO site, almost all the particles were SOA and AS mixtures (> 93%). In the stage 2 samples (0.5-1.0 μm size), SOA and AS mixture particles were dominant for the SA2 and SA3 samples (94% and 82%, respectively). In the stage 2 samples of samples SA1 and SA4, SOA and AS mixture particles were most abundant (50% and 58%, respectively), followed by pure ammonium sulfates (12% and 18%, respectively), aged mineral dust (19% and 8%, respectively), and pure SOA particles (11% and 6%, respectively). In the stage 2 samples of SA1-SA4, the summed contents of SOA and ammonium sulfate were 73%, 99%, 85%, and 82%, respectively, suggesting that SOA and AS are the predominant species in submicron aerosols collected at the ATTO site. The observation of abundant submicron SOAs, which constitute a significant fraction of fine aerosol mass during the wet season at the rainforest, has been reported (Chen et al., 2015; Gilardoni et al., 2011). SOA particles are formed through atmospheric oxidation, condensation, and the gas-to-particle conversion of biogenic volatile organic compounds, such as isoprene and terpenes emitted from the rainforest (Pöschl et al., 2010; Gilardoni et al., 2011; Andreae et al., 2015; Andreae et al., 2018). Although ammonium sulfates were reported to be present in significant quantities in the Amazon basin (Andreae et al., 2015; Chen et al., 2015; Fraund et al., 2017), this study emphasizes the observation of the predominant submicron ammonium sulfates mixed with SOA.

In the stage 2 samples, aged mineral dust and sea-salts (19% and 4%) for the SA1 sample, reacted sea salts (11%) for the SA3 sample, and aged mineral dust and sea-salts (8% and 5%) for the SA4 sample were observed, suggesting that the samples from outside the Amazon rainforest



have different influences because the mineral dust and sea-salt particles were all aged ones. The
influences from outside were observed more clearly for supermicron aerosols at the ATTO site. In
the stage 3 samples (1.0-2.0 μm size) of SA1-SA4, reacted sea-salt particles (33%, 34%, 30%, and
36%, respectively) and aged mineral dust particles (37%, 11%, 7%, and 10%, respectively) were
abundantly observed although the summed relative abundances of SOA and ammonium sulfate
were 28%, 43%, 58%, and 50%, respectively, indicating that SOA and ammonium sulfate are
abundant even in supermicron ATTO aerosols. As all the mineral dust particles were aged, they are
not of local origin, and the observation of a high content (37%) of aged mineral dust particles in
sample SA1 highlights the importance of long-range transatlantic transport (see the fast-moving
air-masses from the Atlantic Ocean for the SA1 sample (Fig. 2(a)). Many studies have examined
the influence of the Saharan dust particles over the Amazon rainforest region, starting with the
measurements made during the ABLE-2B campaign (Talbot et al., 1990; Swap et al., 1992).
Mineral dust is imported most frequently to the rainforest in March and April (Martin et al., 2010a;
Moran-Zuloaga et al., 2018), which increases the ground-based soil dust element levels
significantly (Artaxo et al., 2013). The dust particles can act as ice nuclei and facilitate cold rain
formation (Pöschl et al., 2010). Volatile organic compounds and secondary organics can also
condense on the sea-salt and mineral particles, and the secondary organic coating on the mineral
dust and sea-salt particles can modify their optical, chemical, and hygroscopic properties
significantly (Pöschl et al., 2010; Geng et al., 2014). For example, soluble organic material
coatings on the dust particles may reduce their original ice-nucleating ability and enhance their
cloud-nucleating ability (Tobo et al., 2010; Manktelow et al., 2010). In this study, approximately
70% and 90% of the reacted sea-salt/mineral dust particles in the ATTO and Manaus samples,
respectively, were either mixed or coated with organic matter in the Amazon basin and/or during
the long-range transport.

PBA particles, which are from the rainforest, are sometimes observed in the stage 3 samples

(2%, 10%, 3%, and 3% for the SA1-SA4 samples, respectively). In the stage 4 sample (2.0-4.0 μm
size) of the SA4 sample, the most abundant particle type was PBA (46%), followed in order by
reacted sea-salt (28%), SOA (12%), aged mineral dust (9%), and ammonium sulfate (4%), where
both PBA, SOA, and ammonium sulfate particles of a local origin and the reacted sea-salt and
mineral dust particles from the outside are considerably present. In summary, the aerosols collected
at the ATTO site are mostly SOA and ammonium sulfate of a local origin in the submicron fraction,



although some of the submicron sulfate are of marine and distant origin, whereas aerosols of both local and distant origins are significant in the supermicron fraction.

The aerosols collected at the Manaus site were diverse compared to those at the ATTO site. As shown in Fig. 13, for the stage 1 samples, SOA and ammonium sulfate particles, including their mixture, were the major components for the SM1 and SM2 samples (66% and 62%, respectively), whereas they were only 17% for the SM3 sample. For the stage 2 samples, they were 80%, 53%, and 25% for the SM1-SM3 samples, respectively. As SOA and ammonium sulfate particles can be from the surrounding rainforest areas in addition to local anthropogenic sources, samples SM1-SM3 collected at the Manaus site appear to be influenced from the outside in the order of samples SM1 > SM2 > SM3. In addition, considerable amounts of submicron carbonaceous particles were observed, such as soot, char, and tar balls, which are of a local origin (for the SM1-SM3 samples, 21%, 25%, and 78% for stage 1 and 6%, 29%, and 49% for stage 2, respectively). The most abundant submicron aerosols for sample SM3 were carbonaceous ones, indicating that the local influence to the samples is in the order of SM3 > SM2 > SM1.

In supermicron Manaus aerosols, PBA particles, aged mineral dust, and reacted sea-salts in addition to SOA and carbonaceous particles are abundant. In the stage 3 aerosols of the SM1 sample, the most abundant particles were reacted sea-salts, followed by aged mineral dust, SOA, ammonium sulfate, and PBA particles, which also indicates the strong influence on the SM1 sample from the outside. In stage 3 aerosols of the SM2 sample, the most abundant particles were SOA, followed in order by aged mineral dust, reacted sea-salt, PBA, carbonaceous particles, and ammonium sulfate, which also indicates the strong influence on the SM2 sample from the surrounding rainforest areas. In the stages 3 and 4 aerosols of the SM3 sample, the most abundant particles were aged mineral dust (36% and 52% for stages 3 and 4, respectively), followed by carbonaceous particles, PBA, SOA, and reacted sea-salt. As the aged sea-salt contents were relatively low (10% and 4% for stages 3 and 4, respectively), most of the aged mineral dusts appear to be of a local origin. Fly ash and heavy metal-containing particles of an anthropogenic local origin were considerable in the Manaus samples, i.e., they were observed in the range of 3-7% relative abundances in the Manaus samples. In particular, soot particles of an anthropogenic origin were observed ubiquitously in all the Manaus samples. Among the samples, the aerosols collected at the Manaus site were different. The SM1 sample was influenced most strongly from the outside, including the surrounding rainforest and transatlantic transport. The SM2 sample has some





influences by local sources as well as from the outside. The SM3 sample contains mainly aerosols of an anthropogenic local origin in the submicron fraction and some influences from the outside in the supermicron fraction. Figures 2(e)-(f) show that the backward trajectories at heights of 100-m and 500-m are further from the outside in the order of SM1 > SM2 > SM3, which is consistent with the characteristics of submicron and supermicron aerosols of the SM1-SM3 samples.

**Conclusions**

In this study, aerosol samples collected in the Amazonian rainforest and Manaus, Brazil during the 2012 wet season were investigated on a single particle basis using low-$Z$ particle EPMA. The aerosol particles were classified into nine particle types based on their morphology on SEIs together with the elemental concentrations of a total of 3,162 individual particles: (i) secondary organic aerosols (SOA), (ii) ammonium sulfate (AS) particles, (iii) SOA and AS mixture particles, (iv) aged mineral dust, (v) reacted sea-salts, (vi) primary biological aerosol (PBA) particles, (vii) carbon-rich or elemental carbon (EC) particles such as soot, tar ball, and char, (viii) fly ash particles, and (ix) heavy metal (HM, such as Fe, Zn, Ni, and Ti)-containing particles. For submicron aerosols collected at the ATTO site, the SOA and AS mixture particles were predominant (50-94% in relative abundance) with the summed contents of SOA and ammonium sulfate being 73-100%. In contrast, in the supermicron aerosols at the ATTO site, aged mineral dust and sea-salt (37-70%) as well as SOA and ammonium sulfate (28-58%) were abundant. PBAs were observed abundantly in the $PM_{2-4}$ fraction (46%), and EC and fly ash particles were absent in all the fractions. An analysis of a bulk $PM_{0.25-0.5}$ aerosol sample collected at the ATTO site using RMS and ATR-FTIR showed that ammonium sulfate, organics, and minerals are the major chemical species, which is consistent with the EPMA results.

In the submicron aerosols collected in Manaus, either SOA and ammonium sulfate (17-80%) or EC particles (6-78%) were dominant, depending on the samples. The supermicron aerosols collected in Manaus consisted mainly of aged mineral dust, reacted sea-salts, PBA, SOA, ammonium sulfate, and EC particles. SOA, ammonium sulfate, and PBAs were mostly of a biogenic origin and EC and HM-containing particles were of an anthropogenic origin. The aged mineral dust and reacted sea-salt particles, including mineral dust mixed with sea-salts probably during long-range transatlantic transport, were abundant in the supermicron fractions at both sites. The submicron aerosol at the ATTO site was influenced mainly by the emission from the rainforest



and its supermicron aerosols showed additional contributions from long-range transport, including the Atlantic Ocean and Sahara desert, whereas the aerosols collected in Manaus showed different local and outside contributions among the samples. Among all the aged mineral dust and reacted sea-salt particles, sulfate-containing ones outnumbered those containing nitrates and both nitrate and sulfate in the ATTO samples, whereas N and S containing particles were comparable to sulfate-only ones in the Manaus samples, indicating the different sources and formation mechanisms of secondary aerosols, i.e., the predominant presence of sulfate at the ATTO site from biogenic emissions and elevated influences of nitrates from anthropogenic activities at the Manaus site.

**Author contributions**

LW, RHMG, and CR designed the experiment and LW, XL, and HKK carried out the measurements and analyzed the data. CGGB, CIY, RAFDS, and CP organized and performed the samplings. LW, HG, RHMG, CGGB, AFLG, and CR interpreted the observations and LW, HG, RHMG, CGGB, MOA, and CR drafted the paper.

**Acknowledgement**

This study was supported by Basic Science Research Programs through the National Research Foundation of Korea (NRF) funded by the Ministry of Education, Science, and Technology (NRF-2018R1A2A1A05023254). The work of M. O. Andreae and C. Pöhlker was supported by the German Max Planck Society (MPG). For the operation of the ATTO site, we acknowledge the support by the German Federal Ministry of Education and Research (BMBF contract 01LB1001A), the Brazilian Ministério da Ciência, Tecnologia e Inovação (MCTI/FINEP contract 01.11.01248.00), and the Coordination for the Improvement of Higher Education Personnel (CAPES) for scholarship funding (investigation ) as well as the Amazon State University (UEA), FAPEAM, LBA/INPA, and SDS/CEUC/RDS-Uatumã. A special thanks to Claudomiro Mauricio da Silva for the support during sampling.

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



Figure 1. Location of sampling sites at the Brazilian Amazon basin: an urban site in Manaus (S 3°05.753' W 59°59.419') and a rainforest site at ATTO (S 02°647' W 58°59.992'). Map of South America (top left) with the region marked with a red rectangle and a map of the Amazonas state, Brazil (bottom left) also with the region of interest marked in red.

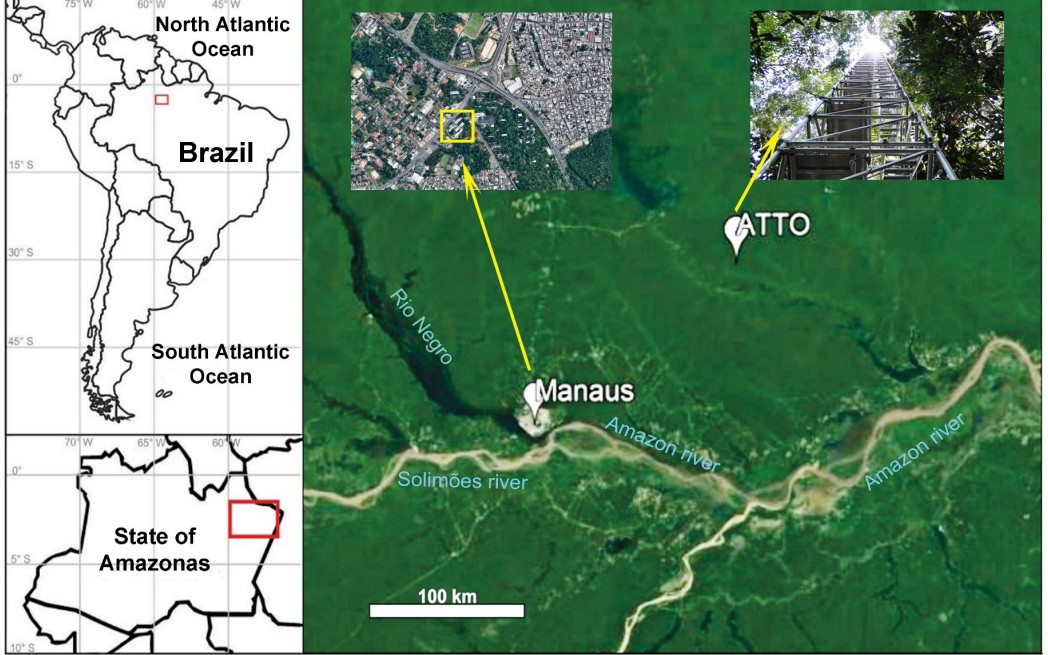





Figure 2. Ten-day (240 h) backward air mass trajectories at 100 m-, 500 m-, and 1000 m-receptor heights; (a)-(d) for the SA1-SA4 samples collected on April 1 and 16-18, 2012 at the ATTO site and (e)-(g) for the SM1-SM3 samples collected on May 1-3, 2012 at the Manaus site. HYbrid Lagrangian Single-Particle Integrated Trajectory (HYSPLIT) model available at the NOAA Air Resources Laboratory's web server (http://www.arl.noaa.gov/ready/hysplit4.html) was used



Figure 3. Home-made sample holder for TEM grid samples in SEM/EDX measurements and a typical X-ray spectrum of the TEM grids.

(a) TEM grids holder          (b) X-ray spectrum of TEM grids

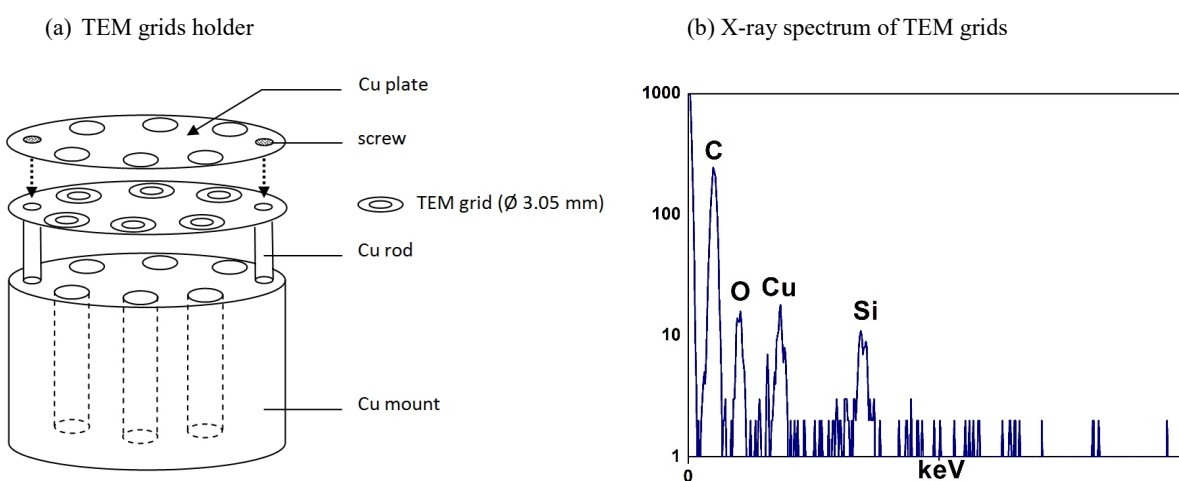



Figure 4. Typical SEM images of aerosol particles for (a) stage 1 (PM$_{0.25-0.5}$) of the SA2 sample, (b) stage 2 (PM$_{0.5-1.0}$) of the SA4 sample, (c) stage 3 (PM$_{1-2}$) of the SA1 sample, and (d) stage 4 (PM$_{2.0-4.0}$) of the SA4 sample, collected at the ATTO site. For convenience, ammonium sulfate, secondary organic aerosol, aluminosilicates, and reacted sea-salt are denoted as "AS", "SOA", "AlSi" and "rss", respectively.

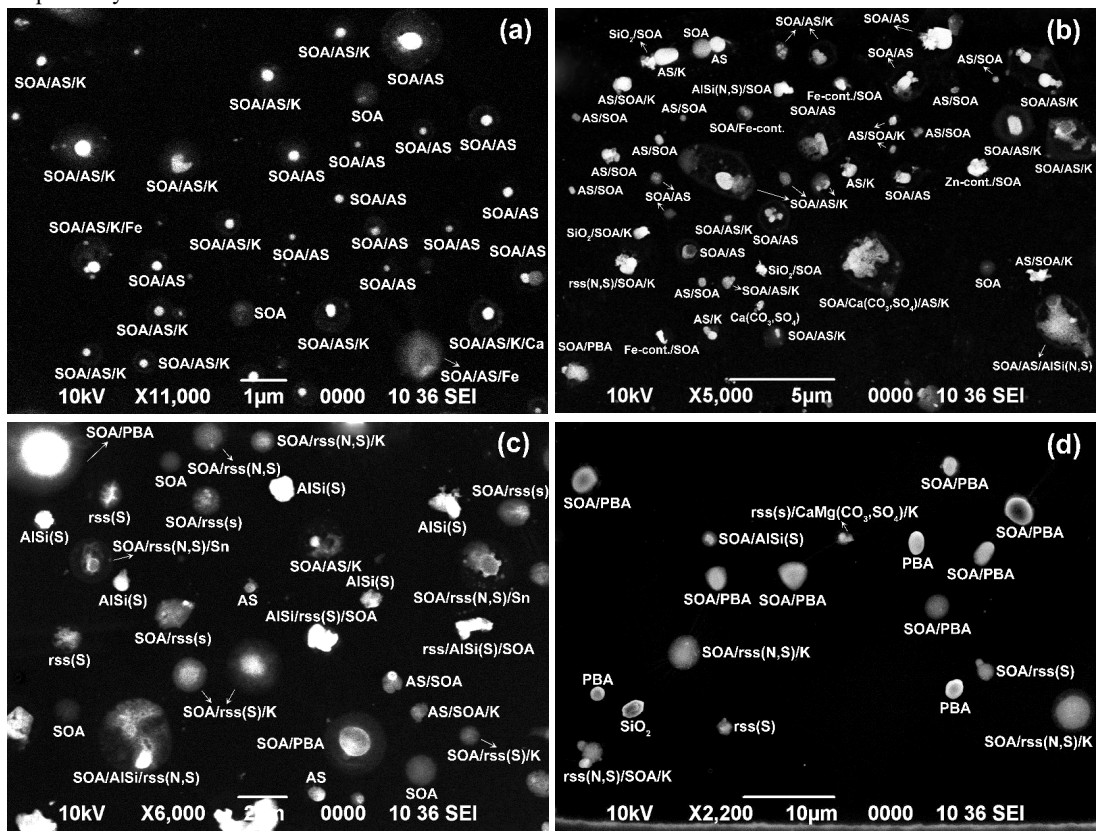




Figure 5. Typical SEM images of aerosol particles for (a) stage 1 ($PM_{0.25-0.5}$) of the SM1 sample, (b) stage 2 ($PM_{0.5-1.0}$) of the SM2 sample, (c) stage 3 ($PM_{1-2}$) of the SM1 sample, and (d) stage 4 ($PM_{2.0-4.0}$) of the SM3 sample, collected at the Manaus site. For convenience, ammonium sulfate, secondary organic aerosol, aluminosilicates, and reacted sea-salt are denoted as "AS", "SOA", "AlSi" and "rss", respectively.

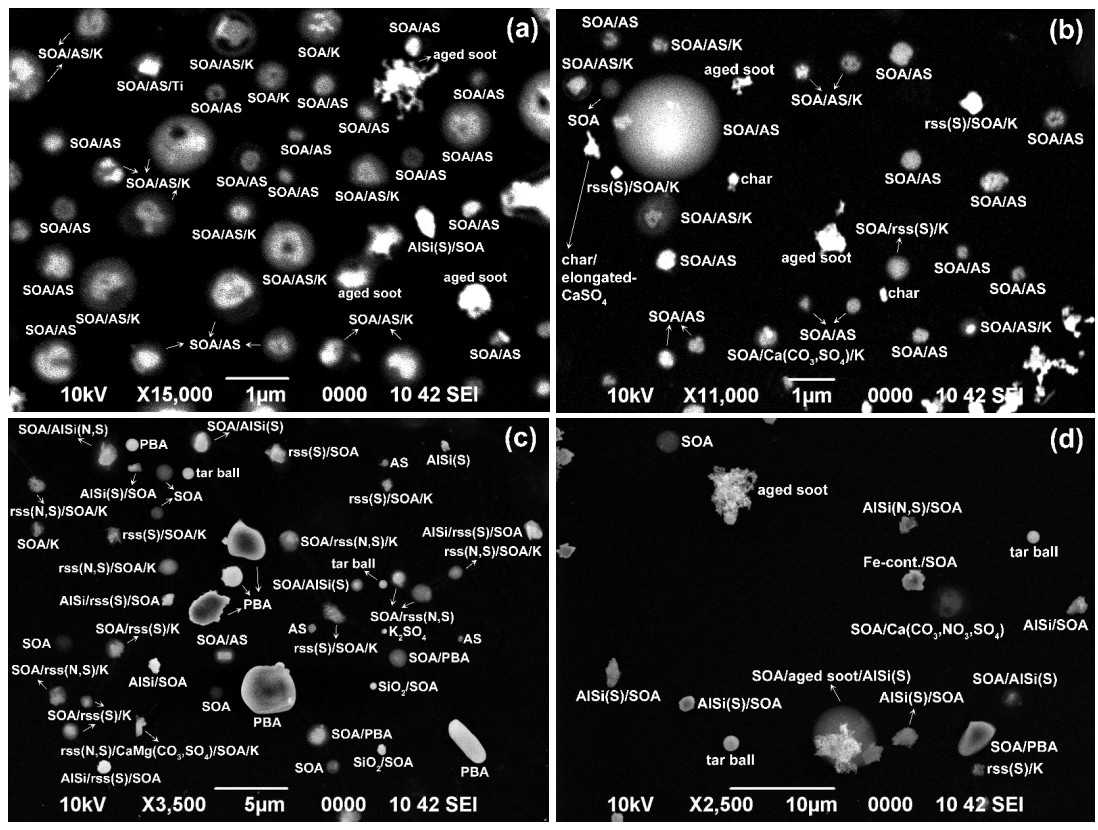



Figure 6. SEIs, X-ray spectra, and element atomic concentrations of SOA, ammonium sulfate (AS), and mixture particles. The inset images in (b), (c), and (f) show the beam damage on the particles after X-ray measurements.

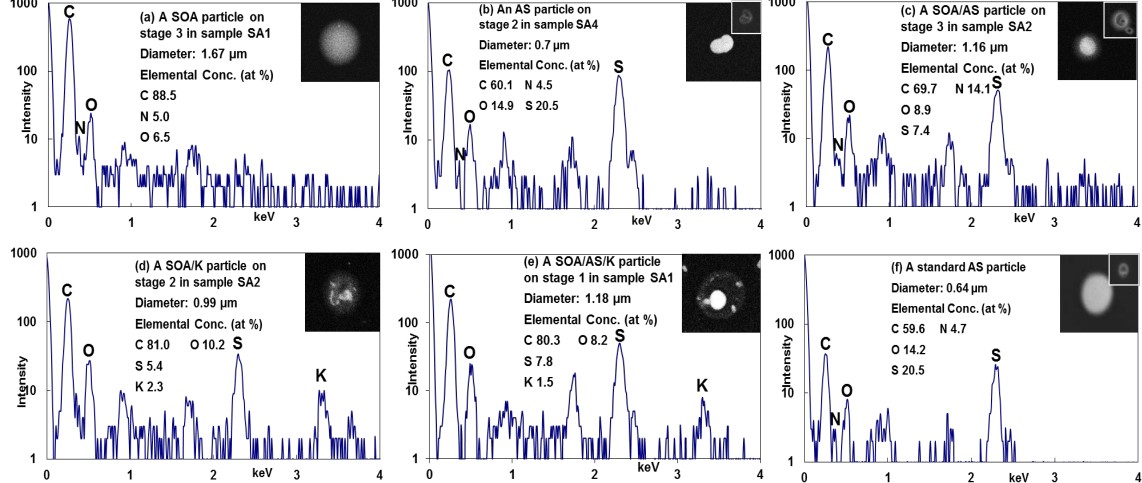





Figure 7. (a) Raman spectra of standard and airborne ammonium sulfate (AS) particles, which were rescaled for clarity. The inset SEI images are for standard and airborne AS particles where the scale bar is 1 μm. The shoulder peak of $SO_4^{2-}$ at 982 cm⁻¹ in the airborne AS particles is from $K_2SO_4$; (b) SEI, optical images, X-ray, ATR-FTIR, and Raman spectra of an overloaded $PM_{0.25-0.5}$ sample collected at ATTO site on June 10, 2014. X-ray, ATR-FTIR, and Raman spectra indicate that AS, organics, and minerals are the major components of the submicron sample. The sloping baseline in the airborne Raman spectrum is due to the fluorescence from organic compounds.

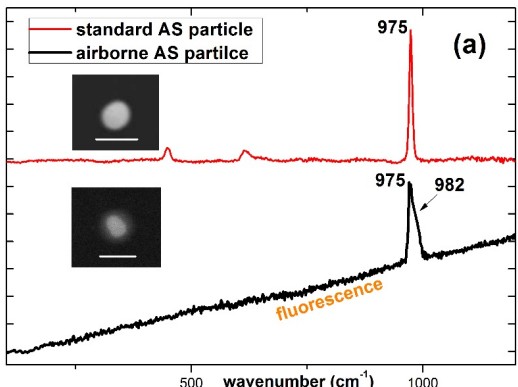

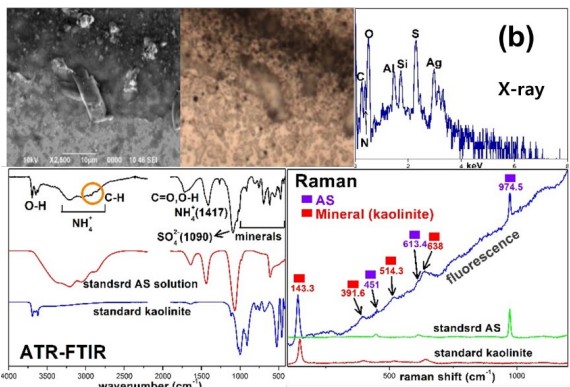



Figure 8. SEIs, X-ray spectra, and element atomic concentrations of aged mineral dust particles.

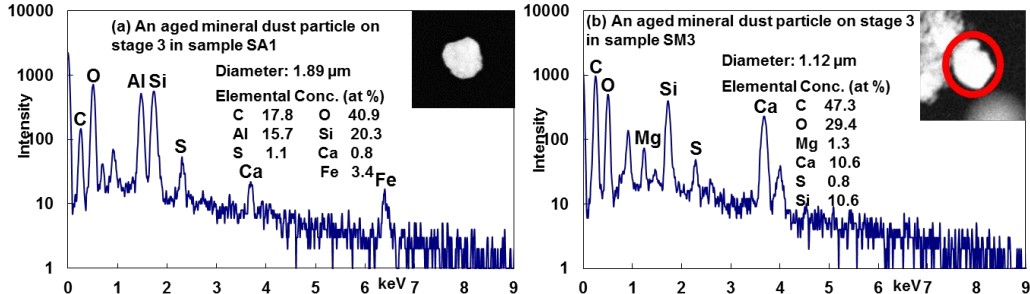



Figure 9. Number abundances of reacted sea-salt and aged mineral dust particles containing sulfates ( ■ ), nitrates ( ■ ),  and both ( ■ ).

(a)  aged mineral dust

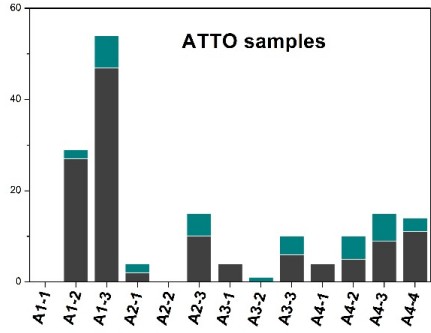
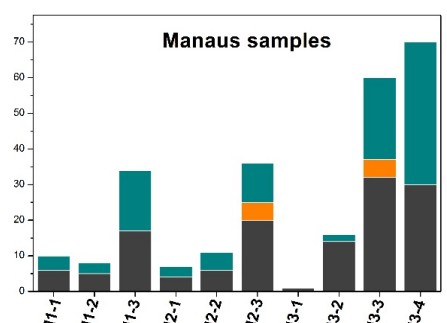

(b)  reacted sea-salt

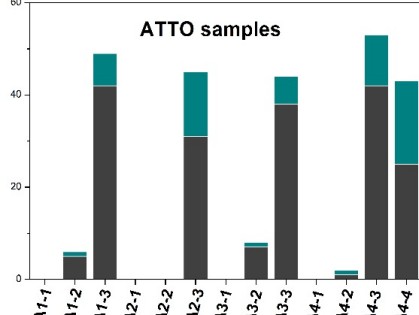
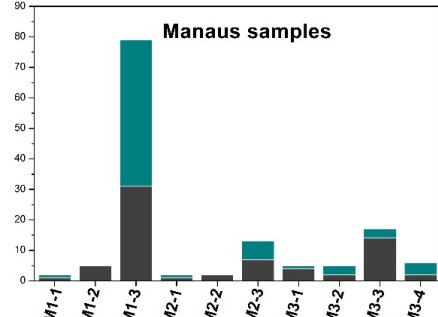





Figure 10. SEIs, X-ray spectra, and element atomic concentrations of (a) reacted sea-salt, (b) reacted sea-salt with K-salt, and (c) elongated $CaSO_4$

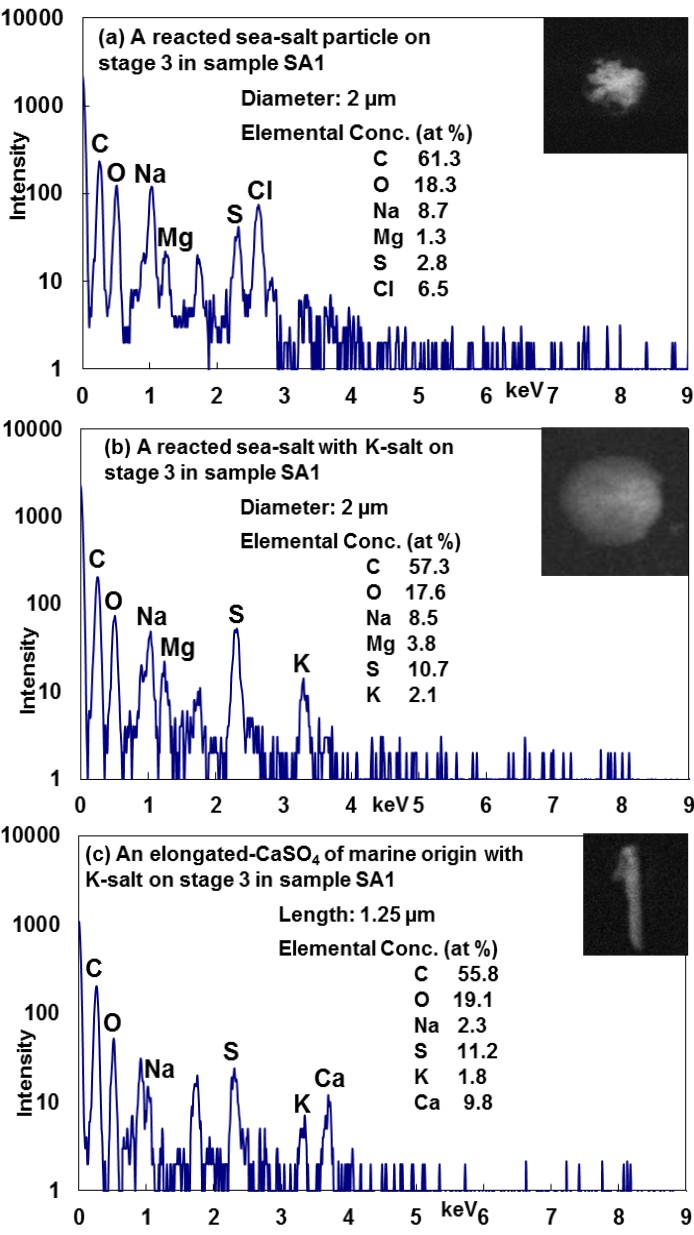





Figure 11. SEIs, X-ray spectra, and element atomic concentrations of (a) PBA, (b) soot, (c) tar call, (d) fly ash, (e) Ni-containing, and (f) Fe-containing particles.

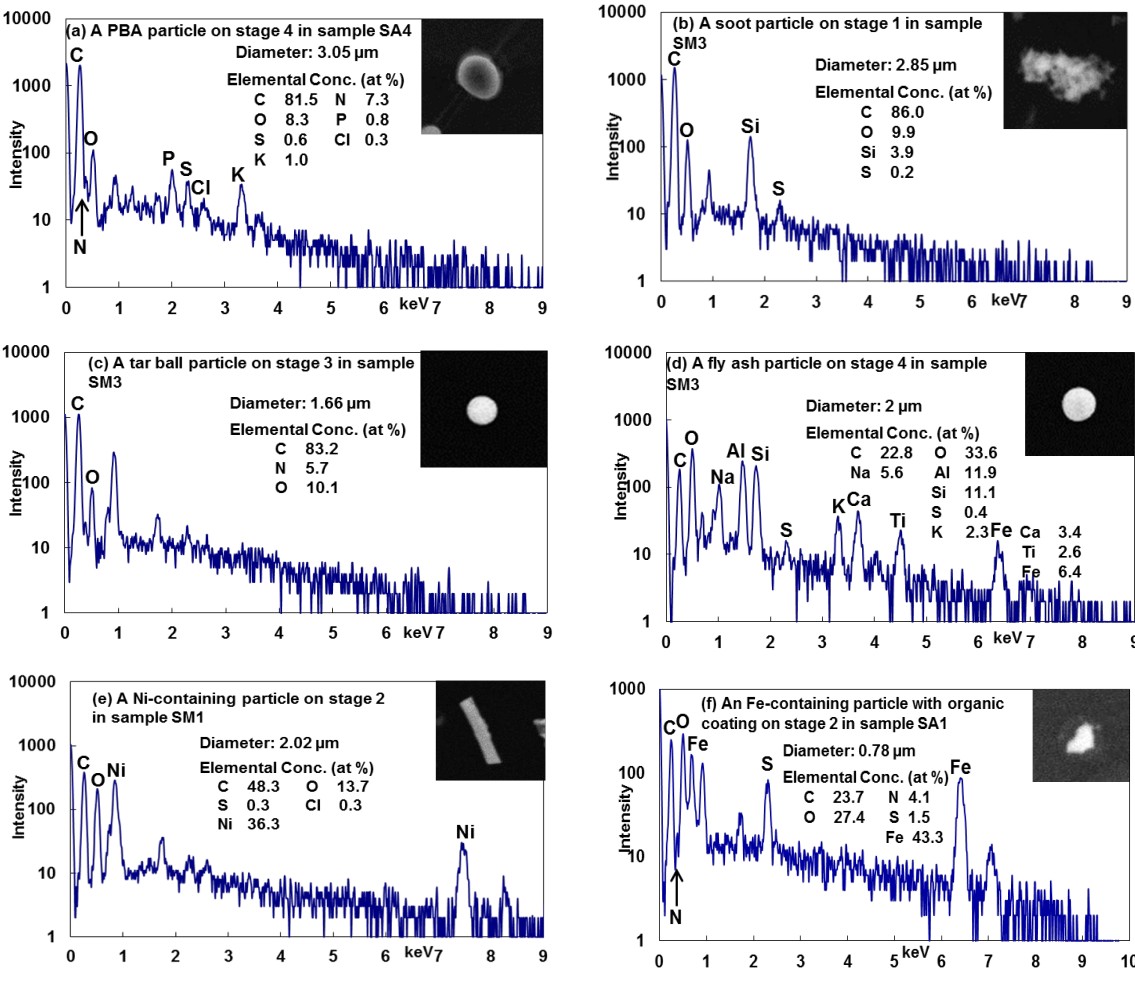




Figure 12. Typical SEIs of PBA particles from stage 4 of the (a) SA4 and (b) SM3 samples. PBA and PBA/SOA mixture particles are marked with (→) and (+), respectively.

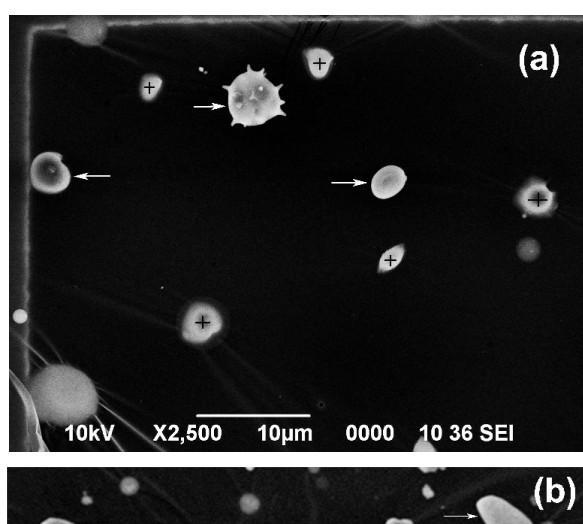

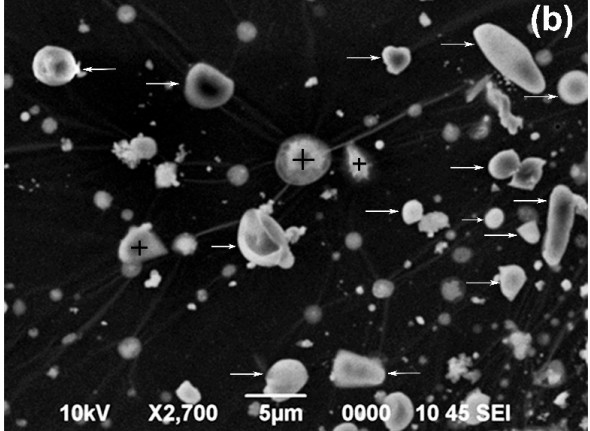





Figure 13. Relative abundance of nine different particle types for the SA1-SA4 and SM1-SM3 samples collected at the ATTO and Manaus sites, respectively.

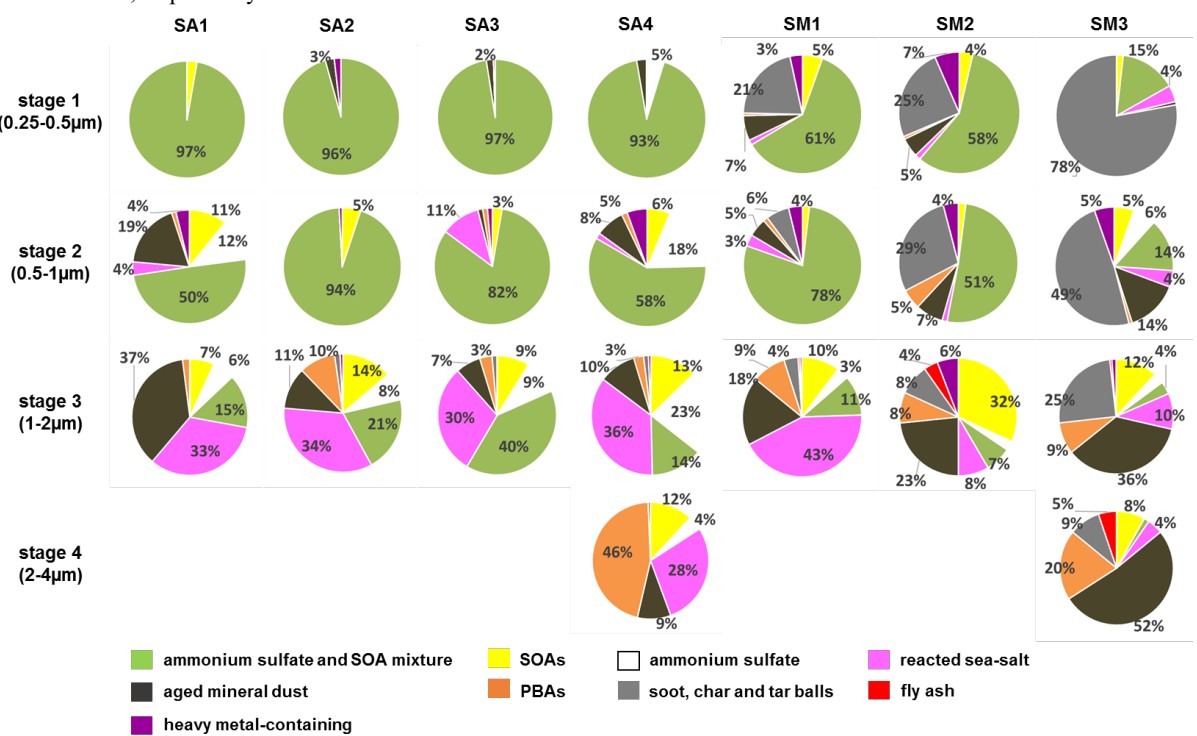