# Peer review of "Manuscript under review for journal Atmos. Chem. Phys."

_Atmospheric Chemistry and Physics, 2018_

## Referee Comment (RC1) · Anonymous Referee #1 · 17 Dec 2018

General comment: the manuscript reports single-particle analysis from size-selected aerosol samples collected at two sites in the Amazon basin, ATTO forest site and Manaus city center, in the wet season of 2012. Results from the forest site samples confirm previous observations reported in the literature, with a predominance of SOA particles in the smaller size ranges. Results from the Manaus samples attest the influence of local urban emissions, as well as the influence of regional biogenic emissions and sea salt. In my opinion, the discussion of the results from Manaus, which is the novelty brought by this manuscript, could be improved. The manuscript is reasonably well written, but there are some redundancies that could be omitted.

[Figure]

Comments to the text:

1) General comment on the introduction: I suggest restructuring the order of the contents. First, present general characteristics of the Amazon region and its importance, mentioning the ATTO site and the Manaus city. Then, focus on aerosol sources in the wet season, and previous studies on single-particle characterization in Amazonia.

2) Line 65: interaction with? Maybe it would be better to replace "interaction" with "dynamical processes".

3) Line 65: aerosols are airborne by definition, so I suggest omitting "airborne" here.

4) Lines 182-188: These sentences, near the beginning of the results section, refer to a previous study, right? It might create a confusion between your current results and previous studies. Please reformulate, to make clear what your contribution is.

5) Line 351: Replace "nascent airborne" by "fresh".

6) Line 377: Replace "size" by "diameter".

7) Line 394: there is a typo here (90%).

8) Lines 439-445: It is not necessary to describe the figure and its numbers in words, like here. I suggest that you go straight to the interpretation of the figure results.

9) Section 3.2: I recommend that you omit redundancies in this section. You already discussed the particle formation processes and its contribution to CCN and IN, there is no need to repeat it. For example, in lines 450-453 and 474-480.

Specific comments:

1) Lines 77-78: There is not a unique pathway to produce SOA in Amazonia. Condensation of low volatility organic species can also occur at the surface of PBA, for example.

2) Lines 84-87: This is true for biomass burning aerosols in Amazonia. This sentence does not fit well into this paragraph, since you are talking mostly about biogenic aerosols.

3) Lines 97-98: This sentence is unclear. First, you might say that Fraund et al. (2017) used cluster analysis to identity particle types. Second, what do you mean by "anthropogenic elements"?

4) Lines 107-108: One third of what? Mass? Number? PM or OOA (oxygenated organic aerosols)? Please clarify.

5) Lines 109-116: Compared to previous works, I would say that the main novelty of this study is the analysis of aerosol samples from the Manaus city center. Emphasize that.

6) Line 127: Please provide more details about the city sampling site. Was it located in the southern on northern part of the city? What was the approximate distance to busy roads/avenues/streets? Do you know if vehicular types were mostly light duty, heavy duty, or both? Is there potential influence of industrial or power plant emissions? The Manaus urban plume is very heterogeneous. You can find more information at de Sá et al., 2018 and Medeiros et al., 2017.

7) Line 132: information on sampling dates and sample names would be better suited in a table.

8) Line 135: How were the samples conserved during transport and storage? Semivolatile organic compounds can volatilize when subjected to temperature variations. Do you think that this is an important issue in your measurements?

9) Line 141: You must provide more information about the backtrajectories calculation. What was the input meteorological database that you used? In which spatial resolution? How many trajectories were calculated for each aerosol sample?

10) Lines 263-264: Is this true (ammonium sulfate dominant over ammonium bisulfate) both for Manaus and ATTO? Is it possible to provide a percentage of the relative ammounts of sulfate in each form (ammonium sulfate, ammonium bisulfate, sulfuric acid)? How does it compare to other metropolis in the world?

11) Line 274: I recommend citing Saturno et al. (2018), about the influence of African volcanic emissions on sulfate concentrations at the ATTO site.

12) Lines 287-209: You discussed the potential sources of SO4 and NH4 in the Amazon forest, based on previous studies. How about the potential sources in Manaus city? You may refer to Medeiros et al., 2017 and de Sá et al., 2018. Also, you may refer to the literature produced on this subject for other metropolis in the world.

13) Lines 343-345: Here you could hypothetize on the potential sources of nitrate in Manaus.

14) Line 364-366: Yes, biomass burning emissions reduce significantly in the Amazonian wet season. However, advection from Africa brings biomass burning aerosols, besides Saharan dust, to the Amazon region in March-April. See, for example, Baars et al., 2011. So, part of the observed K-salts could be associated with biomass burning aerosols from Africa.

15) Line 405: You may cite a reference for the term "brown carbon". It could be either Andreae et al., 2006 or Laskin et al., 2010.

16) Lines 467-469: A single backtrajectory is unsufficient to prove the occurrence of African aerosol advection. You may add a sentence recognizing that.

17) Lines 497-499: The reasoning is not strong. Part of the SOA observed at Manaus can be locally produced. You would need a biogenic tracer to make this point. A better indication of the influence of forest emissions in Manaus is the 20% of PBA in sample SM3, stage 4, provided that the sampling site was not in a city park. The influence of the forest over the city is an interesting question, and I suggest that you include a sentence about it in the abstract.

18) In general, I recommend that you improve the discussion on Manaus results. It is

a metropolis in the middle of a tropical forest, a very unique situation. How does your findings compare to other metropolis in the world? What is the proportion of SOA, PBA and soot in other cities?

19) Lines 502-503: Was there anything special during the sampling SM3? Did you notice increased traffic activity compared to the other days?

20) Line 650: Please update the reference, since it moved from ACPD to ACP.

21) Figure 9: I suggest that you group the columns by size range, for a better comprehension of the plots. Also, you might include information on particle size ranges, either in the x-axis label, or in the figure caption.

References:

Andreae, M. O. and Gelencsér, A.: Black carbon or brown carbon? The nature of light-absorbing carbonaceous aerosols, Atmos. Chem. Phys., 6(3), 3131–3148, 2006.

Baars, H., Ansmann, A., Althausen, D., Engelmann, R., Artaxo, P., Pauliquevis, T. and Souza, R.: Further evidence for significant smoke transport from Africa to Amazonia, Geophys. Res. Lett., 38(20), 1–6, doi:10.1029/2011GL049200, 2011.

Laskin, A., Laskin, J. and Nizkorodov, S. a.: Chemistry of Atmospheric Brown Carbon, Chem. Rev., 115(10), 4335–4382, doi:10.1021/cr5006167, 2015.

Medeiros, A. S. S., Calderaro, G., Guimarães, P. C., Magalhaes, M. R., Morais, M. V. B., Rafee, S. A. A., Ribeiro, I. O., Andreoli, R. V., Martins, J. A., Martins, L. D., Martin, S. T. and Souza, R. A. F.: Power plant fuel switching and air quality in a tropical, forested environment, Atmos. Chem. Phys., 17(14), 8987–8998, doi:10.5194/acp-17-8987-2017, 2017.

Sá, S. S., Palm, B. B., Campuzano-Jost, P., Day, D. A., Hu, W., Isaacman-VanWertz, G., Yee, L. D., Brito, J., Carbone, S., Ribeiro, I. O., Cirino, G. G., Liu, Y., Thalman, R., Sedlacek, A., Funk, A., Schumacher, C., Shilling, J. E., Schneider, J., Artaxo, P.,

[Figure]

Goldstein, A. H., Souza, R. A. F., Wang, J., McKinney, K. A., Barbosa, H., Lizabeth Alexander, M., Jimenez, J. L. and Martin, S. T.: Urban influence on the concentration and composition of submicron particulate matter in central Amazonia, Atmos. Chem. Phys., 18(16), 12185–12206, doi:10.5194/acp-18-12185-2018, 2018.

Saturno, J., Ditas, F., Penning de Vries, M., Holanda, B. A., Pöhlker, M. L., Carbone, S., Walter, D., Bobrowski, N., Brito, J., Chi, X., Gutmann, A., Hrabe de Angelis, I., Machado, L. A. T., Moran-Zuloaga, D., Rüdiger, J., Schneider, J., Schulz, C., Wang, Q., Wendisch, M., Artaxo, P., Wagner, T., Pöschl, U., Andreae, M. O. and Pöhlker, C.: African volcanic emissions influencing atmospheric aerosols over the Amazon rain forest, Atmos. Chem. Phys., 18(14), 10391–10405, doi:10.5194/acp-18-10391-2018, 2018.

---

## Referee Comment (RC2) · Anonymous Referee #2 · 19 Dec 2018

General comments: the manuscript provides information of individual particles in the Amazon basin using a combination of microscopy and infrared techniques, which is an unprecedented approach to the problem. This is the main achievement of the study from the analytic viewpoint. With respect to the location, particles are from the pristine ATTO Tower and urban pollution from the big city of Manaus. If ATTO has been extensively reported in the scientific literature, it is not the case for the city of Manaus, so the manuscript comes to contribute to the knowledge with respect to the properties of aerosols from Manaus.

Specific comments: this referee does agree to all comments posted by the other re-

viewer with respect to explore better the Manaus database. I would add to this the poor discussion with respect to the the weak point of the manuscript is with respect to a better of the atmospheric condition as a whole. For example, the characterization of the meteorology (lines 138 - 144) was extremelly poor. Nothing was written on the synoptic situation. Hysplit is useful but it does not consider wet removal adequately thus a 10 day back trajectory is far from enough to provide a good information about meteorology. To say that RH was above 55% in the Amazon is useless, it is almost all the time from January - May above it. So, were the measured days ordinary? Anything different?

With respect to the discussion about the emissions from Manaus, only 9 lines (100 - 108) were written about it with very few information. So, the city is industrial but the author did not mention that these industries are not great emitters, they are basically assembling industries. The main source of pollutants is the light vehicles fleet. Poor information was also provided with respect to the sampling site in Manaus. Also there were a mention about downwind transport of pollution from Manaus that confuses the reader (lines 104-108) because ATTO is upwind and barely get any influence from the city. The focus of this is far from any downwind issues.

Comments to the text: the text is very well written. I would add the following corrections:

tar ball > tarball (several locations in the text). Figures 6, 8, 10, 11: fix the weird position of the axis label "kev". Figure 9: put dates instead of sample label in X-axis. Figure 11 (caption): tar call > tarball. Figure 13 (caption): "Relative abundance" to "Relative MASS abundance"

---

## Author Comment (AC1) · 18 Jan 2019

**General Comment from Anonymous Referee #1**

*The manuscript reports single-particle analysis from size-selected aerosol samples collected at two sites in the Amazon basin, ATTO forest site and Manaus city center, in the wet season of 2012. Results from the forest site samples confirm previous observations reported in the literature, with a predominance of SOA particles in the smaller size ranges. Results from the Manaus samples attest the influence of local urban emissions, as well as the influence of regional biogenic emissions and sea salt. In my opinion, the discussion of the results from Manaus, which is the novelty brought by this manuscript, could be improved. The manuscript is reasonably well written, but there are some redundancies that could be omitted.*

**Response:** We thank the reviewer for his/her positive evaluation of our work and valuable comments. We revised the manuscript as much as possible respecting the reviewer's comments.

**Comments to the text from Anonymous Referee #1**

*1) General comment on the introduction: I suggest restructuring the order of the contents. First, present general characteristics of the Amazon region and its importance, mentioning the ATTO site and the Manaus city. Then, focus on aerosol sources in the wet season, and previous studies on single-particle characterization in Amazonia.*

**Response:** As suggested, the introduction was revised (see the revised introduction).

*2) Line 65: interaction with? Maybe it would be better to replace "interaction" with "dynamical processes".*

**Response:** Done as suggested.

*3) Line 65: aerosols are airborne by definition, so I suggest omitting "airborne" here.*

**Response:** Done as suggested.

*4) Lines 182-188: These sentences, near the beginning of the results section, refer to a previous study, right? It might create a confusion between your current results and previous studies. Please reformulate, to make clear what your contribution is.*

**Response:** The sentences were modified mainly by inserting the phrase, "In a previous study (Pöschl, et al., 2010)", to avoid the possible confusion. The sentence, "With the exception of the reacted sea-salt particles probably from the Atlantic Ocean as well as the abundant ammonium sulfate aerosols, the particle types observed in this study are comparable to their study", was given.

*5) Line 351: Replace "nascent airborne" by "fresh".*

**Response:** Done as suggested.

*6) Line 377: Replace "size" by "diameter".*

**Response:** Done as suggested.

*7) Line 394: there is a typo here (90%).*

**Response:** Done as suggested.

*8) Lines 439-445: It is not necessary to describe the figure and its numbers in words, like here. I suggest that you go straight to the interpretation of the figure results.*

**Response:** Revised as suggested.

*9) Section 3.2: I recommend that you omit redundancies in this section. You already discussed the particle formation processes and its contribution to CCN and IN, there is no need to repeat it. For example, in lines 450-453 and 474-480.*

**Response:** As recommended, the redundancies were omitted.

**Specific Comments from Anonymous Referee #1**

*1) Lines 77-78: There is not a unique pathway to produce SOA in Amazonia. Condensation of low volatility organic species can also occur at the surface of PBA, for example.*

**Response:** In order to avoid such impression, a "mainly" word was added – "SOA particles are mainly formed through the condensation of biogenic organic compounds onto biogenic K-rich salt particles emitted from the forest".

*2) Lines 84-87: This is true for biomass burning aerosols in Amazonia. This sentence does not fit well into this paragraph, since you are talking mostly about biogenic aerosols.*

**Response:** The reviewer's comment is right. This part is deleted without deteriorating the context.

*3) Lines 97-98: This sentence is unclear. First, you might say that Fraund et al. (2017) used cluster analysis to identity particle types. Second, what do you mean by "anthropogenic elements"?*

**Response:** The reviewer is right as the sentence is not clear for people who are not familiar with the paper. Without deteriorating the context, the sentence was omitted.

*4) Lines 107-108: One third of what? Mass? Number? PM or OOA (oxygenated organic aerosols)? Please clarify.*

**Response:** Thank the reviewer for the comment. The sentence was modified as "where one third of the potential SOA would be of an urban origin".

*5) Lines 109-116: Compared to previous works, I would say that the main novelty of this study is the analysis of aerosol samples from the Manaus city center. Emphasize that.*

**Response:** As suggested, the sentence, "Especially, single-particle characterization of aerosols collected at a Manaus city center has been scarce." was added.

*6) Line 127: Please provide more details about the city sampling site. Was it located in the southern on northern part of the city? What was the approximate distance to busy roads/avenues/streets? Do you know if vehicular types were mostly light duty, heavy duty, or both? Is there potential influence of industrial or power plant emissions? The Manaus urban plume is very heterogeneous. You can find more information at de Sá et al., 2018 and Medeiros et al., 2017.*

**Response:** More detailed description is given – "In Manaus, the sampling site is situated in the central part of the city (S 03° 05.753', W 59° 59.419'), which is at a representative urban region influenced by electricity production based on fuel oil, diesel, and natural gas, biogenic emissions from the surrounding forest, and mostly by light duty (using gasoline and ethanol) vehicle traffic. Heavy vehicles that use diesel account to less than 10 % of the urban fleet (Medeiros et al., 2017). The location is nearby a small parking area and around 200 m away from the intersection of four busy avenues, with frequent diurnal traffic jam on weekdays.".

*7) Line 132: information on sampling dates and sample names would be better suited in a table.*

**Response:** As suggested by the reviewer, information on samples and meteorological parameters (temp., RH, rainfall, and meteorological conditions during the samplings) were given in Table 1.

*8) Line 135: How were the samples conserved during transport and storage? Semivolatile organic compounds can volatilize when subjected to temperature variations. Do you think that this is an important issue in your measurements?*

**Response:** During the transport of storage of the samples, semivolatile organic compounds can volatilize as the reviewer pointed out. However, more critically, the particles are under high vacuum during the EPMA measurements, which makes the detection of semivolatile components impractical. The EPMA measurements are not for semivolatile components in aerosols.

*9) Line 141: You must provide more information about the backtrajectories calculation. What was the input meteorological database that you used? In which spatial resolution? How many trajectories were calculated for each aerosol sample?*

**Response:** As suggested, a sentence was added – "In the HYSPLIT calculation, meteorological data output from the Global Data Assimilation System (GDAS) using GDAS1 data with a horizontal resolution of 1° corresponding to ~ 100 km × 100 km and 23 vertical layers was used, which was reported to provide plausible backward trajectory analysis (Su et al., 2015).".

*10) Lines 263-264: Is this true (ammonium sulfate dominant over ammonium bisulfate) both for Manaus and ATTO? Is it possible to provide a percentage of the relative amounts of sulfate in each form (ammonium sulfate, ammonium bisulfate, sulfuric acid)? How does it compare to other metropolis in the world?*

**Response:** Raman spectroscopy used in this work detected the ammonium sulfate in the samples, but no ammonium bisulfate and sulfuric acid. The Raman analysis was for the qualitative analysis rather than the quantitative one.

*11) Line 274: I recommend citing Saturno et al. (2018), about the influence of African volcanic emissions on sulfate concentrations at the ATTO site.*

**Response:** As suggested by the reviewer, Saturno et al. (2018) was cited – "including that from the African volcanic emissions (Saturno et al., 2018)".

*12) Lines 287-289: You discussed the potential sources of SO4 and NH4 in the Amazon forest, based on previous studies. How about the potential sources in Manaus city? You may refer to Medeiros et al., 2017 and de Sá et al., 2018. Also, you may refer to the literature produced on this subject for other metropolis in the world.*

**Response:** A sentence was added for major ammonium sulfate sources in the urban environment – "In the urban environment, anthropogenic ammonium sulfate is mainly formed by gaseous reactions among $SO_2$ emitted from coal-fired plants and industrial activities, $NH_3$ emitted from human and animal activities and fertilization in the fields, and oxidants (e.g., $O_3$ and OH radical) (Li et al., 2016; Geng et al., 2017).".

*13) Lines 343-345: Here you could hypothetize on the potential sources of nitrate in Manaus.*

**Response:** As suggested, a phrase was added – "The nitrates for the Manaus samples may be formed from nitrogen oxides emitted from the vehicles and coal-fired power plants (Geng et al., 2014, 2017; Li et al., 2016).".

*14) Line 364-366: Yes, biomass burning emissions reduce significantly in the Amazonian wet season. However, advection from Africa brings biomass burning aerosols, besides Saharan dust, to the Amazon region in March-April. See, for example, Baars et al., 2011. So, part of the observed K-salts could be associated with biomass burning aerosols from Africa.*

**Response:** Thank the reviewer for the comment. A phrase was added – "although it was reported that a strong biomass burning smoke was transported from Africa to South America during the wet season (Baars et al., 2011) so that a part of the observed K-salts could be associated with biomass burning aerosols from Africa.".

*15) Line 405: You may cite a reference for the term "brown carbon". It could be either Andreae et al., 2006 or Laskin et al., 2010.*

**Response:** Done as suggested.

*16) Lines 467-469: A single backtrajectory is unsufficient to prove the occurrence of African aerosol advection. You may add a sentence recognizing that.*

**Response:** As suggested, a phrase was added – "although a single backtrajectory cannot confirm the African origin of the aged mineral dust particles.".

*17) Lines 497-499: The reasoning is not strong. Part of the SOA observed at Manaus can be locally produced. You would need a biogenic tracer to make this point. A better indication of the influence of forest emissions in Manaus is the 20% of PBA in sample SM3, stage 4, provided that the sampling site was not in a city park. The influence of the forest over the city is an interesting question, and I suggest that you include a sentence about it in the abstract.*

**Response:** It was stated in the original manuscript that a part of SOA is from local sources. However, the SOA and AS contents of samples SM1 and SM2 are clearly higher than that of sample SM3, so that the reasoning can be defendable. A sentence suggested by the reviewer was added in the abstract – "Based on the different contents of SOA, ammonium sulfate, and EC particles among the samples collected in Manaus, a considerable influence of the rainforest over the city was observed.".

*18) In general, I recommend that you improve the discussion on Manaus results. It is a metropolis in the middle of a tropical forest, a very unique situation. How does your findings compare to other metropolis in the world? What is the proportion of SOA, PBA and soot in other cities?*

**Response:** I agree with the reviewer in that Manaus is a very unique place. However, it is not feasible to compare the current results with those in other metropolis in the world. We could not find literature data of SOA, PBA, and soot in other cities collected in the similar way as this work. I would like to mention that this single-particle characterization of aerosols has some difficulty in comparing with the results from bulk analyses.

*19) Lines 502-503: Was there anything special during the sampling SM3? Did you notice increased traffic activity compared to the other days?*

**Response:** Indeed, only sample SM3 was collected during a regular working day, which is given in the revised text – "Samples SM1 and SM2 were collected during and just after a national holiday, respectively, when all the institutions (private and public) were closed during that day so that the traffics were quite low, similar to a weekend or vacation period. Sample SM3 was collected during a regular working day, so that sample SM3 is the only sample actually exposed to the high traffic of light vehicles in the area.".

*20) Line 650: Please update the reference, since it moved from ACPD to ACP.*

**Response:** Done as suggested.

*21) Figure 9: I suggest that you group the columns by size range, for a better comprehension of the plots. Also, you might include information on particle size ranges, either in the x-axis label, or in the figure caption.*

    **Response:** Done as suggested.

---

## Author Comment (AC2) · 18 Jan 2019

**General comments from Anonymous Referee #2**

*The manuscript provides information of individual particles in the Amazon basin using a combination of microscopy and infrared techniques, which is an unprecedented approach to the problem. This is the main achievement of the study from the analytic viewpoint. With respect to the location, particles are from the pristine ATTO Tower and urban pollution from the big city of Manaus. If ATTO has been extensively reported in the scientific literature, it is not the case for the city of Manaus, so the manuscript comes to contribute to the knowledge with respect to the properties of aerosols from Manaus.*

**Response:** We thank the reviewer for the positive evaluation of our work.

**Specific comments from Anonymous Referee #2**

*This referee does agree to all comments posted by the other reviewer with respect to explore better the Manaus database. I would add to this the poor discussion with respect to the weak point of the manuscript is with respect to a better of the atmospheric condition as a whole. For example, the characterization of the meteorology (lines 138 - 144) was extremely poor. Nothing was written on the synoptic situation. Hysplit is useful but it does not consider wet removal adequately thus a 10 day back trajectory is far from enough to provide a good information about meteorology. To say that RH was above 55% in the Amazon is useless, it is almost all the time from January - May above it. So, were the measured days ordinary? Anything different?*

**Response:** Detailed meteorological conditions were added in Table 1. Based on the meteorological parameters during the samplings, the sampling days do not seem abnormal.

*With respect to the discussion about the emissions from Manaus, only 9 lines (100 - 108) were written about it with very few information. So, the city is industrial but the author did not mention that these industries are not great emitters, they are basically assembling industries. The main source of pollutants is the light vehicles fleet. Poor information was also provided with respect to the sampling site in Manaus. Also there were a mention about downwind transport of pollution from Manaus that confuses the reader (lines 104-108) because ATTO is upwind and barely get any influence from the city. The focus of this is far from any downwind issues.*

**Response:** The reviewer is correct as the industries in the region do not emit a large amount of pollutants, whereas power plants, refinery, and vehicle fleet are mainly responsible for the atmospheric emissions and the vehicle fleet is the main source. Light vehicles powered by gasoline, ethanol, or both account to the majority of the transportation fleet in the city. The information was added to the text.
As suggested by the reviewer, the confusing downwind-upwind part was deleted – "Based on an investigation on particulate matter during the wet season, oxidized organic components were significantly observed at Manaus sites (de Sá et al., 2018)".

**Comments to the text from Anonymous Referee #2**

*The text is very well written. I would add the following corrections:*
*1. tar ball > tarball (several locations in the text).*
*2. Figures 6, 8, 10, 11: fix the weird position of the axis label "keV".*
*3. Figure 9: put dates instead of sample label in X-axis.*
*4. Figure 11 (caption): tar call > tarball.*
*5. Figure 13 (caption): "Relative abundance" to "Relative MASS abundance"*

**Response:** Done as suggested, except (i) comment #3, for which Fig. 9 was modified as suggested by Reviewer #1 and (ii) comment #5, for which *"Relative abundance" was replaced by "Relative number abundance"*.